# Evaluation of Pharmacokinetic Feasibility of Febuxostat/L-pyroglutamic Acid Cocrystals in Rats and Mice

**DOI:** 10.3390/pharmaceutics15082167

**Published:** 2023-08-21

**Authors:** Jeong-Eun Yu, Byoung Hoon You, Mingoo Bae, Seung Yon Han, Kiwon Jung, Young Hee Choi

**Affiliations:** 1College of Pharmacy and Integrated Research Institute for Drug Development, Dongguk University_Seoul, 32 Dongguk-ro, Ilsandong-gu, Goyang-si 10326, Gyeonggi-do, Republic of Korea; yyy0982@naver.com (J.-E.Y.); hoon4131@nate.com (B.H.Y.); nophra88@naver.com (M.B.); hsyglory@gmail.com (S.Y.H.); 2College of Pharmacy, CHA University, 335 Pangyo-ro, Bundang-gu, Seongnam-si 13488, Gyeonggi-do, Republic of Korea; 3Oncobix Co., Ltd., 120 Heungdeokjungang-ro, Giheung-gu, Yongin-si 16950, Gyeonggi-do, Republic of Korea

**Keywords:** febuxostat, FBX/L-pyroglutamic acid cocrystals, pharmacokinetics, bioavailability

## Abstract

Febuxostat (FBX), a selective xanthine oxidase inhibitor, belongs to BCS class II, showing low solubility and high permeability with a moderate *F* value (<49%). Recently, FBX/L-pyroglutamic acid cocrystal (FBX-PG) was developed with an improving 4-fold increase of FBX solubility. Nevertheless, the in vivo pharmacokinetic properties of FBX-PG have not been evaluated yet. Therefore, the pharmacokinetic feasibility of FBX in FBX- and FBX-PG-treated rats and mice was compared in this study. The results showed that the bioavailability (*F*) values of FBX were 210% and 159% in FBX-PG-treated rats and mice, respectively. The 2.10-fold greater total area under the plasma concentration–time curve from time zero to infinity (AUC_0-inf_) of FBX was due to the increased absorption [i.e., 2.60-fold higher the first peak plasma concentration (*C*_max,1_) at 15 min] and entero-hepatic circulation of FBX [i.e., 1.68-fold higher the second peak plasma concentration (*C*_max,2_) at 600 min] in FBX-PG-treated rats compared to the FBX-treated rats. The 1.59-fold greater AUC_0-inf_ of FBX was due to a 1.65-fold higher *C*_max,1_ at 5 min, and a 1.15-fold higher *C*_max,2_ at 720 min of FBX in FBX-PG-treated mice compared to those in FBX-treated mice. FBX was highly distributed in the liver, stomach, small intestine, and lungs in both groups of mice, and the FBX distributions to the liver and lungs were increased in FBX-PG-treated mice compared to FBX-treated mice. The results suggest the FBX-PG has a suitable pharmacokinetic profile of FBX for improving its oral *F* value.

## 1. Introduction

Pharmaceutical cocrystals are defined as crystalline molecular complexes of fixed stoichiometry between an active pharmaceutical ingredient (API) with other pharmaceutically acceptable molecules [1,2,3]. Cocrystals can improve the physicochemical properties (e.g., stability, solubility, dissolution rate, and/or permeability) of compounds [4,5,6,7,8] and can also be tuned with pharmaceutically acceptable co-formers without altering the chemical structure of the API [1]. Especially, cocrystals are known to be superior to conventional formulations in in vivo pharmacokinetic properties such as absorption and bioavailability (*F*) [9,10,11].

Febuxostat (FBX) has been popularly used to treat hyperuricemia, gout, and other chronic renal diseases [12,13]. FBX reduces uric acid levels in the blood by selectively inhibiting xanthine oxidase [14,15,16]. From physicochemical and pharmacokinetic perspectives, FBX is a poorly water-soluble and highly permeable drug belonging to the biopharmaceutics classification system (BCS) class II [1,17,18,19,20,21]. Thus, various formulations of FBX, such as salt formations, solid dispersion formations, and cocrystals, have been developed to improve the solubility of FBX [22,23,24,25]. Cocrystals of FBX have especially gained attention. Although FBX form-A is still used in drug formulation, the co-crystallization of FBX was widely attempted due to the low aqueous solubility and slow absorption rate of FBX as a restriction of BCS class II drugs [1,14,18,26,27,28,29,30,31,32,33]. Many cases to enhance the solubility of FBX were reported in numerous FBX cocrystals with isonicotinamide, arginine, pyridine, 4-hydroxy benzoic acid, or salicylic acid as a co-former [22,24,34], but these cocrystals of FBX exhibited unstable properties in the buffered solutions of pH 4–7 [34].

Among various trials to develop cocrystals of FBX, L-pyroglutamic acid (PG) was appropriate because of its safety (i.e., LD_50_ of PG is 1 g/kg in rats [35]) and its chemical properties (i.e., a pKa difference of less than 3 between FBX and PG [18,36]). The cocrystals of FBX were formed with FBX/PG at a ratio of 2:1, leading to enhanced solubility in deionized water and simulated gastrointestinal pH conditions (pH 1.2, 4.0, and 6.8) [18]. Considering that the solubility and stability of an orally administered drug in the different pH conditions of the gastrointestinal tract especially influence the absorption and bioavailability [37], FBX/PG cocrystals (FBX-PG) have an advantage due to the enhanced thermodynamical stability and solubility of FBX-PG in both deionized water and buffers, reflecting the pH conditions of the gastrointestinal tract [18]. However, the in vivo pharmacokinetic properties of FBX after the administration of FBX-PG has not been evaluated yet. Therefore, the pharmacokinetic feasibility of FBX-PG, a novel cocrystal of FBX, was compared with that of FBX in rats and mice.

## 2. Materials and Methods

### 2.1. Materials

FBX (purity 98.00%) and FBX-PG (99.85%) (Figure 1), with identical spectroscopic characteristics, were provided by Dr. Kiwon Jung at the College of Pharmacy, CHA University and Oncobix Co., Ltd. (Yongin-si, Republic of Korea) [18,38]. A brief summary of the manufacturing process of FBX-PG [18,38] is as follows: the solid powder of FBX and the organic solvent (methanol or acetone) are mixed, and PG is added. Thereafter, the temperature (20–25 °C) is raised and refluxed, and then the refluxed reaction solution is cooled and stirred. Finally, FBX-PG is obtained by vacuum drying. Carbamazepine [internal standard (IS) for liquid chromatography-tandem mass spectrometry (LC-MS/MS)] was purchased from Sigma-Aldrich (St. Louis, MO, USA). All other chemicals and reagents used were of analytical grade.

### 2.2. Animals

The protocols for the animal studies were approved by the Institute of Laboratory Animal Resources of Dongguk University, Seoul, Republic of Korea (IACUC-2016-023 on 18 March 2016, and IACUC-2022-012 on 10 June 2022). Male Sprague Dawley rats (6–7 weeks old, weighing 150–250 g) and male Institute of Cancer Research mice (6 weeks old, weighing 25–30 g) were purchased from the Charles River Company Korea (Orient, Seoul, Republic of Korea). The animals were acclimated for one week before starting the study. Upon arrival, the animals were randomized and housed at two (for rats) or five (for mice) per cage under strictly controlled environmental conditions (20–25 °C and 48–52% relative humidity). A 12 h light/dark cycle was used at an intensity of 150 to 300 Lux. Animals had free access to food and water. They were fasted for 12 h before drug administration.

### 2.3. LC-MS/MS Analysis of FBX in the Biological Samples

The concentration of FBX in the biological samples was determined using a previously reported LC-MS/MS method with a slight modification [16]. All analyses were performed using an API 4000 triple quadrupole mass spectrometer (AB Sciex, Foster City, CA, USA) coupled with an Agilent 1200 high-performance liquid chromatography system (Agilent, Santa Clara, CA, USA).

The mass spectrometer was operated in the multiple reaction monitoring mode with an electrospray ionization interface. For positive ions ([M+H]^+^), the source temperature and gas parameters were optimized as follows: ion spray voltage was set to 5500 V, turbo ion spray temperature was set at 500 °C, nebulizer gas (GS1, nitrogen) and heater gas (GS2, nitrogen) were set at 50 L/min, curtain gas was set at 20 L/min, and collision gas (nitrogen) was set at 6 Torr. The *m*/*z* values of 317.115 → 261 (29 eV for collision energy) and 237.241 → 194.3 (52 eV for collision energy) were obtained for FBX and the IS, respectively.

Chromatographic separation was carried out using a reversed-phase C_18_ column (X-select C_18_, 2.1 mm × 100 mm i.d., particle size; 3 μm; Waters, Dublin, Ireland) at a flow rate of 0.4 mL/min. The mobile phase was composed of 0.1% formic acid in water (A) and acetonitrile (B). Gradient elution was performed using the mobile phase at a 90:10 (*v*/*v*) ratio of A:B initially. The ratio was changed to 10:90 (*v*/*v*) from 0.5 min to 3.5 min and was returned to the initial composition at 7.1 min, which was then maintained for 10 min. The analytical data were processed using Analyst software (Version 1.7.2, AB Sciex, Foster City, CA, USA).

A stock solution of FBX was prepared at a concentration of 14 mg/mL in methanol to make FBX standards in the biological samples. The stock solution was serially diluted with methanol. Different concentrations of FBX working solutions were added to the drug-free plasma, urine, or gastrointestinal tract (GI) samples to prepare standard samples of FBX in biological samples.

For sample preparation, a 50 µL aliquot of sample was deproteinized by adding 950 µL of acetonitrile containing 1 µg/mL of IS. After vortex and centrifuging for 10 min at 13,523× *g*, a 5 µL aliquot of the supernatant was obtained and diluted with acetonitrile containing 0.1% formic acid, resulting in a 25,000-fold dilution of the biological samples. Then, a 10 µL aliquot of the diluted supernatant was injected into the column. The FBX and IS peaks appeared at 5.5 and 3.8 min, respectively (Appendix A). The concentration range of the FBX calibration curves of plasma, urine, and GI samples was 1.4–140 µg/mL. The calibration curves for FBX were derived from the peak area ratios relative to those of the IS via linear regression with 1/x weighting.

### 2.4. Pharmacokinetics of FBX after Oral Administration of FBX or FBX-PG to Rats

Rats were fasted for 12 h prior to the experiment but were allowed access to water. On the day of the experiment, the rats were randomly divided into two groups, namely, the FBX (*n* = 12) and FBX-PG (*n* = 13) groups, respectively. The rats were exposed to diethyl ether by inhalation for 5 min. Under anesthesia, the carotid artery was cannulated for blood sampling according to a previously reported method [39,40,41]. Upon recovery from the anesthesia (i.e., 3 h after cannulation), FBX or FBX-PG (dissolved in ethanol and polyethylene glycol at 5:5, *v*/*v*) at a dose of 50 mg (4 mL)/kg as FBX was orally administered to the rats using a gastric gavage tube. To administer the equivalent dose of FBX within FBX-PG, the dose of FBX-PG was adjusted by considering the molar concentration of FBX (316.37 g/mol) and PG (129.04 g/mol) within FBX-PG, considering that FBX-PG (761.85 g/mol) consists of a 1:2 ratio of FBX and PG. Blood samples (approximately 150 µL) were collected via the carotid artery 0, 5, 15, 30, 60, 90, 120, 240, 360, 480, 600, 720, 960, 1200, and 1440 min after the oral administration of FBX or FBX-PG. Immediately after each blood collection, a 0.4 mL of heparinized 0.9% NaCl-injectable solution was injected into the cannula to prevent blood clotting. Each blood sample was immediately centrifuged, and a 50 µL aliquot of plasma was collected. At the end of 24 h, each metabolic cage was rinsed with 20 mL of distilled water, and the resulting fluid was mixed with the urine collected over the previous 24 h and taken as the urine sample. After manual shaking and stirring, a 50 µL aliquot of the urine sample was collected. At the same time, each rat was euthanized by cervical dislocation. The GI tract, including its contents and feces, was removed, transferred into a beaker, and cut into small pieces. To facilitate the extraction of FBX, 100 mL of methanol was added to each beaker. After manual shaking and stirring, 50 µL of supernatant was collected from each beaker. All collected biological samples were stored at −20 °C for LC-MS/MS analysis of FBX.

### 2.5. Pharmacokinetics of FBX after Oral Administration of FBX or FBX-PG to Mice

Prior to the experiment, the mice were fasted for 12 h but were allowed access to water. On the day of the experiment, the mice were randomly divided into two groups, namely, the FBX (*n* = 22) and FBX-PG (*n* = 15) groups, respectively. Anesthesia was conducted via i.p. injection of 0.05 mL per kg, composing a 3:1 mixture of zoletil (i.e., tiletamine 125 mg + zolazepam 125 mg/5 mL) and rompun (xylazine HCl 23.3 mg/5 mL) before heart puncture followed by the previous report [42,43,44]. FBX and FBX-PG (dissolved in ethanol and polyethylene glycol at 5:5, *v*/*v*) were orally administered to the mice at a dose of 50 mg (10 mL)/kg as FBX using a gastric gavage tube. To administer the equivalent dose of FBX within FBX-PG, the dose of FBX-PG was adjusted by considering the molar concentration of FBX and PG within FBX-PG as described in the pharmacokinetic study in rats. Blood samples (approximately 120 µL) were collected via heart puncture using a heparinized insulin syringe at 0, 5, 15, 30, 60, 120, 240, 360, 480, 600, 720, 960, and 1440 min after the oral administration of FBX and FBX-PG. In heart puncture, a 31G needle was employed to minimize damage to cardiac and pericardial tissues along the needle track and to keep the mice alive for several blood collections. Blood samples were immediately centrifuged, and 50 µL of plasma was collected and stored at −20 °C for LC-MS/MS analysis of FBX.

### 2.6. Tissue Distribution of FBX after Oral Administration of FBX or FBX-PG to Mice

The tissue distribution study was performed according to the previously reported methods [45,46]. Prior to the experiment, mice were fasted for 12 h but were allowed access to water. On the experiment day, mice were randomly divided into two groups of mice, namely, the FBX (*n* = 30) and FBX-PG (*n* = 15) groups, respectively. Anesthesia was conducted via i.p. injection of 0.05 mL per kg composing a 3:1 mixture of zoletil (i.e., tiletamine 125 mg + zolazepam 125 mg/5 mL) and rompun (xylazine HCl 23.3 mg/5 mL) before heart puncture, followed by the previous report [42,43,44]. FBX or FBX-PG (dissolved in ethanol and polyethylene glycol at 5:5, *v*/*v*) at a dose of 50 mg (10 mL)/kg as FBX was orally administered to the mice using a gastric gavage tube. To administer the equivalent dose of FBX within FBX-PG, the dose of FBX-PG was adjusted by considering the molar concentration of FBX and PG within FBX-PG as described in the pharmacokinetic study in rats. At 0.5, 2, 4, 8, or 12 h after the oral administration of FBX or FBX-PG, as much blood as possible was collected via heart puncture, and then the portal vein was perfused with 0.9% NaCl solution to remove all blood from the body. Each blood sample was centrifuged for 10 min at 13,523× *g*, and a 50 μL aliquot of the plasma was collected. The liver, kidney, stomach, small intestine, large intestine, lung, heart, and fat were excised, weighed, and washed with cold 0.9% NaCl solution. Each tissue was homogenized in a 3-fold volume of 0.9% NaCl solution, which was centrifugated for 20 min at 7607× *g*, and 50 μL of the supernatant was collected. All collected samples were stored at −20 °C until LC-MS/MS analysis of FBX.

### 2.7. Plasma Protein Binding of FBX and FBX-PG in Rats and Mice

The plasma protein binding values of FBX and FBX-PG in rats were measured at a final concentration equivalent to 1 µg/mL FBX using a rapid equilibrium dialysis (RED) device (Thermo Fisher Scientific, Waltham, MA, USA). A 100 µL sample of fresh rat plasma containing FBX or FBX-PG was added to the plasma chamber, and 350 µL of dialysis buffer solution (phosphate-buffered saline) was added to the buffer chamber. After incubating for 4 h at 37 °C with stirring at 300 rpm, 50 µL was transferred from each chamber into a tube and stored at −20 °C for LC-MS/MS analysis of FBX.

The plasma protein binding values of FBX and FBX-PG in mice were also measured at two concentrations (equivalent to 1 and 20 µg/mL FBX final concentrations). Except for the concentration of FBX or FBX-PG, the other procedures were the same as those used to measure the plasma protein binding values in rats described above.

### 2.8. Pharmacokinetic Analysis

The total area under the plasma concentration–time curve from time zero to the last blood sampling time (AUC_0–1440min_) or infinity (AUC_0-inf_) was calculated using the trapezoidal rule method. The area from the last datum point to infinity was estimated by dividing the last measured plasma concentration by the terminal-phase rate constant.

Standard methods [47] were used to calculate the following pharmacokinetic parameters using non-compartmental analysis (PK solver, version 2.1; Scientific, Sunnyvale, CA, USA): terminal half-life (t_1/2_); apparent time-averaged total body clearance after oral administration (CL/*F*); apparent volume of distribution during elimination after oral administration (V_z_/*F*); relative bioavailability from time zero to the last blood sampling time (relative *F*_0–1440min)_ or to time infinity (relative *F*); percentage of the dose recovered from the gastrointestinal tract (including its contents and feces) at 24 h (GI_24h_); and the percentage of the dose excreted in urine up to 24 h (*Ae*_0–24h_). The relative *F* or *F*_0–1440min_ was estimated by dividing the AUC_0-inf_ or AUC_0–1440min_ after the oral administration of FBX-PG via AUC_0-inf_ or AUC_0–1440min_ after the oral administration of FBX at the same dose. The first peak plasma concentration (*C*_max,1_) and time to reach *C*_max,1_ (*T*_max,1_), and second peak plasma concentration (*C*_max,2_) and time to reach *C*_max,2_ (*T*_max,2_) were directly obtained from the plasma concentration–time data. The mean ‘true’ unabsorbed fractions of the oral dose (‘*F*_unabs_’) were estimated using a reported equation [48]: GI_24h, oral_ = ‘*F*_unabs_’ + (*F* × GI_24h, intravenous_).

### 2.9. Statistical Analysis

The statistical comparison of pharmacokinetic parameters between the two groups was performed using the Student’s *t*-test. A *p*-value of <0.05 was deemed to be statistically significant. All data are expressed as the mean ± standard deviations except for *T*_max,1_ and *T*_max,2_, which are expressed as medians (ranges).

## 3. Results

### 3.1. Pharmacokinetics of FBX after the Oral Administration of FBX or FBX-PG to Rats

The mean plasma concentration–time profiles of FBX after the oral administration of FBX or FBX-PG at a dose of 50 mg (4 mL)/kg as FBX to rats are shown in Figure 2, and the relevant pharmacokinetic parameters of FBX are listed in Table 1. FBX was detected in plasma based on an early blood sampling time point (5 min) for both rats and the time to reach the first peak concentration (*T*_max,1_) value was 30 min and 15 min for FBX and FBX-PG-treated rats, respectively. These results suggest the considerably rapid absorption of FBX from the GI tract in FBX and FBX-PG-treated rats. The first peak plasma concentration (*C*_max,1_) was higher (160% increase) at the shorter *T*_max,1_ (50% decrease) in FBX-PG-treated rats than in FBX rats, indicating that FBX absorption was faster in FBX-PG-treated rats than in FBX rats. A second peak appeared in both FBX and FBX-PG-treated rats, indicating that the biliary excretion of FBX occurred even after FBX-PG administration, similar to the FBX administration reported previously [49,50]. Interestingly, the second peak plasma concentration (*C*_max,2_) was higher (67.7% increase), and the time to reach the second peak concentration (*T*_max,2_) was longer (25.0% increase) in FBX-PG-treated rats than in FBX-treated rats. A smaller GI_24h_ (21.1% decrease) of FBX was observed in FBX-PG-treated rats compared to FBX-treated rats. The *Ae*_0–24h_ of FBX was comparable between FBX and FBX-PG-treated rats, at 3.64% and 3.69%, respectively. Although GI_24h_ is the sum of the unabsorbed and biliary excreted fraction of oral dose at 24 h, the comparable terminal t_1/2_ values of FBX of 434 min and 472 min suggest that FBX elimination was not different between FBX and FBX-PG-treated rats. As a result of drug absorption and disposition, the AUC_0-inf_ of FBX was significantly increased (110% increase) in FBX-PG-treated rats compared to FBX-treated rats. Thus, the relative *F* value of FBX-PG was 210% that of FBX-treated rats.

### 3.2. Pharmacokinetics of FBX after the Oral Administration of FBX or FBX-PG to Mice

The mean plasma concentration–time profiles of FBX after the oral administration of FBX and FBX-PG at a dose of 50 mg (10 mL)/kg to mice are shown in Figure 3, and the relevant pharmacokinetic parameters are listed in Table 2. FBX was detected in plasma based on an early blood sampling time point (5 min) for both groups of mice, and the *T*_max,1_ values were 30.0 min and 5.00 min in FBX and FBX-PG-treated mice, respectively. These results indicated the rapid GI absorption of FBX in FBX and FBX-PG-treated mice. Moreover, the higher *C*_max,1_ (65.0% increase) at the shorter *T*_max,1_ (83.3% decrease) of FBX in FBX-PG-treated mice than FBX-treated mice indicated that the FBX absorption might be faster and greater in FBX-PG-treated mice than in FBX-treated mice. A second peak of FBX appeared in FBX and FBX-PG-treated mice. The *C*_max,2_ at *T*_max,2_ values were 15.3 μg/mL at 480 min and 17.6 μg/mL at 720 min in FBX and FBX-PG-treated mice, respectively. These second peak profiles suggested that biliary excretion might be involved in the elimination of FBX in both FBX and FBX-PG-treated mice. The terminal t_1/2_ values of FBX, at 880 min and 883 min, were similar in FBX and FBX-PG-treated mice, respectively. As a result of absorption and disposition, the AUC_0-inf_ of FBX in FBX-PG-treated mice was greater (58.7% increase) than in FBX-treated mice, with a relative *F* value of 159% in FBX-PG-treated mice compared to FBX-treated mice.

### 3.3. Tissue Distribution of FBX after the Oral Administration of FBX or FBX-PG to Mice

The FBX concentrations in the plasma and various tissues, the ratios of tissue-to-plasma (T/P ratios), and AUC_0–720min_ values at 0.5, 2, 4, 8, and 12 h after the oral administration of FBX or FBX-PG to mice at a dose of 50 mg (10 mL)/kg are shown in Figure 4, Table 3 and Appendix A. In FBX-treated mice, FBX was more highly distributed in the lung, stomach, small intestine, and liver than in other tissues (e.g., large intestine, kidneys, heart, and fat). Although differences in distribution depend on the uptake time to tissues and elimination time from tissues, the T/P ratios of FBX in the lung, stomach, liver, small intestine, large intestine, and kidneys were above 1. FBX distribution in the heart and fat was relatively small compared to other tissues because the T/P ratios in the heart and fat were almost below 1. In FBX-PG-treated mice, the tissue distribution patterns of FBX (i.e., FBX concentrations and its T/P ratios) were similar to those in FBX-treated mice. However, the AUC_0–720min_ values of FBX in the lungs and liver were significantly increased (132% and 216% increase, respectively) in FBX-PG-treated mice compared to FBX-treated mice (Table 3).

### 3.4. Plasma Protein Binding of FBX and FBX-PG in Rats and Mice

The rat plasma protein binding values of FBX in fresh rat plasma spiked with FBX and FBX-PG were 98.4 ± 1.20% and 98.2 ± 1.33% at a final concentration equivalent to 1 μg/mL FBX. The mouse plasma protein binding values of FBX in fresh mouse plasma spiked with FBX and FBX-PG were 94.9 ± 0.0369% and 94.2 ± 1.16% at a final concentration equivalent to 1 μg/mL FBX and 95.7 ± 0.00339% and 95.4 ± 0.143% at a final concentration equivalent to 20 μg/mL FBX.

## 4. Discussion

In a pharmacokinetic evaluation, the concentration–time profiles of a drug in the blood and tissues can explain the absorption rate and systemic and/or specific tissue exposure of the drug. Drug exposure at appropriate concentrations for a sufficient duration is a critical factor in achieving the expected efficacy. Thus, drug exposure to the pharmacokinetic characteristics of a drug play an important role in determining the expected drug response over time (i.e., efficacy and/or toxicity) [51,52]. In this study, the pharmacokinetic feasibility of FBX-PG was characterized by an in vivo system using rodents. Rodents, such as rats and mice, are the preferred species for drug development due to their anatomical, physiological, and genetic similarities to humans. Furthermore, rats and mice have been widely employed in the field of pharmacokinetics as representative models due to their convenience (i.e., small body size and ease of maintenance), physiological resemblance to humans, and abundant genetic resources [53]. Rat models have traditionally been used due to the advantage of allowing multiple blood sampling, which facilitates the calculation of pharmacokinetic parameters [54]. Mouse models have gained attention as appealing alternatives for the following reasons: (1) smaller amounts of compounds are sufficient for pharmacokinetic studies, (2) various disease models have been developed using mice, and/or (3) mice are invaluable for animal-scale up studies [53,55,56]. Based on these advantages, both rat and mouse models were employed in this study to affirm the improved pharmaceutical performance of FBX-PG compared to FBX.

When in vivo pharmacokinetic studies were conducted, it was necessary to adjust the appropriate methodologies, especially for the vehicle and blood collection technique. In choosing the vehicle, the amount of ethanol used in a vehicle, 5:5 (*v*/*v*) of ethanol: polyethylene glycol, was in the safe range [57,58]. The orally administered ethanol volume, 2 mL/kg and 5 mL/kg to rats and mice, respectively, in this study is equivalent to 1.98 g/kg and 3.95 g/kg for rats and mice, considering the density of ethanol as 0.789 g/mL. Based on the reported LD_50_ values of orally administered ethanol, 10.6 g/kg and 12 g/kg in rats and mice [57,59], the safe amount of ethanol was used as a vehicle in this study. In addition, multiple blood samplings should be designed depending on the animals and blood collection technique [43,44]. For the blood collection technique in rats, the blood was collected through catheterization of the carotid artery. A single blood sampling of 0.15 mL and a total blood volume of 3 mL, which closely approximates 10% of the total blood volume per rat, are recommended in the guideline. This study employed a single blood sampling of 0.15 mL at 14 blood sampling time points, and a total blood sampling volume of 2.25 mL, adhering to the reported criteria. In other words, the total blood sampling volume, 2.25 mL in this study, does not exceed the suggested value of 3.0 mL [60]. Furthermore, to prevent excessive blood loss, blood transfusion was administered between 12 h and 24 h blood sampling time points. For the blood collection technique in mice, heart puncture (also known as cardiac puncture) using a syringe equipped with a 19–27G needle is commonly involved [61,62,63]. Guidelines recommend a single blood sampling volume of 0.15 mL, a total blood sampling volume of 1.8 mL, and approximately three times of heart punctures in mice with 20–25 g body weight [42,43,64]. In this study, 0.12 mL of blood was collected at each sampling time three or four times, resulting in 0.36–0.48 mL of total blood being collected in each mouse with 25–30 g of body weight. As each plasma concentration−time profile was constructed using data from three or four mice, blood was collected at three or four time points from each individual mouse. In other words, the total blood sampling volume, 0.36–0.48 mL in this study, does not exceed the suggested value, 1.8 mL [62,65]. Furthermore, a 31G needle was employed to minimize damage to cardiac and pericardial tissues along the needle track, ensuring the mice’s survival for multiple blood collections. Thus, the blood collection procedure adhered to the appropriate criteria for rats and mice was performed.

There are three possible contributing factors for the low *F* of FBX. First, the low solubility of FBX restricts its dissolution and/or absorption processes in the GI tract, resulting in incomplete and variable absorption. This leads to a low oral dose *F* value [18,20,66]. In the evaluation of pharmacokinetic feasibility of FBX and FBX-PG, the higher *C*_max,1_ and shorter *T*_max,1_ of FBX observed after FBX-PG administration (Table 1 for rats and Table 2 for mice) indicated a faster absorption compared to FBX administration. It is worth noting that GI_24h_, which represents the sum fraction of the unabsorbed dose and biliary excreted dose 24 h after oral administration, does not provide an exact calculation of the unabsorbed percentage of the oral dose [48]. Therefore, the mean ‘true’ unabsorbed fraction (‘*F*_unabs_’) after the oral administration of FBX to rats was calculated based on a reported equation.
GI_24h, oral_ = ‘*F*_unabs_’ + (*F* × GI_24h, intravenous_)In FBX-treated rats, the assumed value for GI_24h_ of the intravenous dose (GI_24h, intravenous_) was approximately 8.5%, based on a previously reported value of the dose excreted in the bile after intravenous administration [49]. The absolute *F* of FBX in rats was estimated to be approximately 49% using a previously reported value [13]. In FBX-PG-treated rats, the same value for GI_24h, intravenous_ was utilized, and the estimated absolute *F* was 103%*,* considering the relative *F* of 210% in rats.
0.517 = ‘*F*_unabs_’ + (0.49 × 0.085) → FBX-treated rats
0.408 = ‘*F*_unabs_’ + (1.03 × 0.085) → FBX-PG-treated ratsAccording to An et al. (2017), the solubility of FBX-PG was improved by approximately 3.92 in water and a 1.95–5.19 fold in various pH conditions compared to FBX. This improved solubility of FBX-PG might affect the absorption rate of FBX, which results in higher *C*_max,1_ at shorter *T*_max,1_ in FBX-PG compared to those in FBX. Moreover, the increased solubility of FBX-PG caused the enhanced absorption extent of FBX as well as the absorption rate of FBX, which was supported by the absorbed fraction values of FBX were 52.5 and 68.0% in FBX and FBX-PG-treated rats, respectively. Similar phenomena were already reported in numerous cases: increased solubility of a cocrystal formation of API with a conformer resulted in increased absorption rate and extent of each API, such as hesperetin cocrystal with piperine [67], TAK-020 (a Bruton tyrosine kinase inhibitor) cocrystal with gentisic acid [68], and ketoconazole cocrystal with 4-aminobenzoic acid [69]. Second, another contributing factor to the low *F* is its extensive first-pass effect. After absorption, FBX undergoes significant metabolism and biliary excretion, further reducing its systemic availability [49,50]. FBX is extensively metabolized by cytochrome P450 enzymes (CYP 1A1, 1A2, 2C8, 2C9, and 3A4) as well as uridine 5′-diphospho-glucuronosyltransferase (UGT) [20,50,70]. It has been reported that approximately 20–30% of glucuronidated FBX, such as FBX N-glucuronide, is excreted in the urine, along with less than 10% unchanged FBX [49,50,70,71]. Furthermore, a second peak pattern of FBX itself, attributed to biliary excretion, has also been observed [71]. It was reported that approximately 8.5% of the administered FBX dose was excreted in the bile after intravenous administration to rats [49]. Third, distribution can be considered. FBX exhibits high protein binding capacity (greater than 99%), which can impact its distribution and availability in the systemic circulation [49]. These pharmacokinetic properties of FBX contribute to its low *F* value. In this study, the high plasma protein binding values of FBX were observed in both FBX and FBX-PG-treated rats and mice, ranging from approximately 94.2% to 98.4%. These results indicated that the free fractions of FBX were similarly low in both rodents.

The possible mechanisms for the appearance of a second peak in FBX and FBX-PG-treated rats include enterohepatic circulation, solubility-limited absorption, and reversible distribution between tissues and plasma [72,73,74]. Firstly, the second peak may be attributed to the enterohepatic circulation of FBX in rats, as previously reported [17,71]. The higher *C*_max,2_ with longer *T*_max,2_ in FBX-PG-treated rats compared to FBX-treated rats suggests that biliary-excreted FBX may be more readily reabsorbed into the bloodstream. The similar terminal t_1/2_ of FBX, at 434 min and 472 min, in both FBX and FBX-PG-treated rats, respectively, supports the notion that the elimination of FBX in both groups of rats was comparable. Secondly, solubility-limited absorption, particularly observed in BCS class II drugs with low solubility, can lead to the appearance of a second peak after oral administration. Similar second peak profiles at approximately 4–8 h were reported for levodopa and doxycycline [72]. Thirdly, reversible distribution between tissue and plasma can also contribute to the occurrence of a second peak. Despite its high plasma protein binding value, FBX rapidly distributes to the lung, stomach, and liver, indicating the possibility of the reversible movement of FBX between tissues and plasma over time. Multiple peaks have been observed in drugs, such as trimethoprim and sulfadiazine, due to their extensive distribution and reversible movement between tissues and systemic circulation [75]. As a result of absorption and disposition, systemic exposure (i.e., AUC_0-inf_, *C*_max,1_, and *C*_max,2_), as well as the relative *F* value of FBX, were increased in FBX-PG-treated rats compared to FBX-treated rats (Table 1 and Figure 2). This could be attributed to the enhanced absorption of FBX in the GI tract and its subsequent reabsorption into the bloodstream through enterohepatic circulation. The increased solubility of FBX in FBX-PG may have contributed to this phenomenon. Similar to the observations in rats, increased relative *F* values and increased systemic exposure (i.e., increased AUC_0-inf_, *C*_max,1_, and *C*_max,2_) of FBX were observed in FBX-PG-treated mice compared to FBX-treated mice (Table 2 and Figure 3). And, similar to rats, a second peak was also observed in the mice. FBX-PG-treated mice exhibited higher *C*_max,1_ at shorter *T*_max,1_, and higher *C*_max,2_ at longer *T*_max,2_ compared to FBX-treated mice. During tissue distribution, mice were only employed due to the limited quantity of FBX-PG available for tissue distribution studies. FBX initially distributes to the stomach and small intestine during absorption and then extensively distributes to highly perfused organs, such as the liver and lungs. In FBX-treated mice, FBX showed higher distribution in the stomach, liver, and lungs compared to other tissues. The distribution pattern may vary depending on the extraction time of the tissue, which reflects the distribution process. T/P ratios of FBX above 1 were observed in the stomach, small intestine, large intestine, lungs, and liver, indicating the preferential accumulation of FBX in these tissues. In contrast, the distribution in the heart and fat tissue was slower and smaller, with T/P ratios below 1. In FBX-PG-treated mice, the tissue distribution of FBX, including FBX concentrations and T/P ratios, appeared to be similar to those in FBX-treated mice. However, the distribution of FBX in the liver and lungs seemed to be increased in FBX-PG-treated mice compared to FBX-treated mice. Additionally, the T/P ratios in almost all tissues increased from 0.5 to 8 h, even though the plasma concentrations of FBX decreased after 2 h. This suggests that FBX continues to be distributed to tissues even when its concentrations in the bloodstream decrease. The findings indicate that FBX was well distributed to various tissues with high affinity, and the distribution pattern of FBX was largely comparable between FBX and FBX-PG-treated mice. The differential rates of FBX distribution in various tissues can provide valuable information for selecting suitable in vivo disease models to further evaluate the in vitro potency of FBX [76].

## 5. Conclusions

It can be concluded that the FBX-PG administration significantly increased the AUC_0-inf_ and *F* values of FBX in rats and mice. Despite the high plasma protein binding value of FBX, the tissue distribution of FBX to livers and lungs exhibited significant increases after FBX-PG administration compared to FBX. These findings suggest that the potential of FBX-PG as a promising formulation to enhance FBX exposure in both plasma and tissues, thereby possibly improving therapeutic efficacy. However, further studies are required to elucidate the changes between exposure and therapeutic efficacy of FBX after FBX-PG administration. Additionally, exploring underlying mechanisms, such as entero-hepatic circulation, provide valuable insights to explain the systemic exposure of FBX beyond the increased absorption observed with oral administration of FBX-PG.

## Figures and Tables

**Figure 1 pharmaceutics-15-02167-f001:**
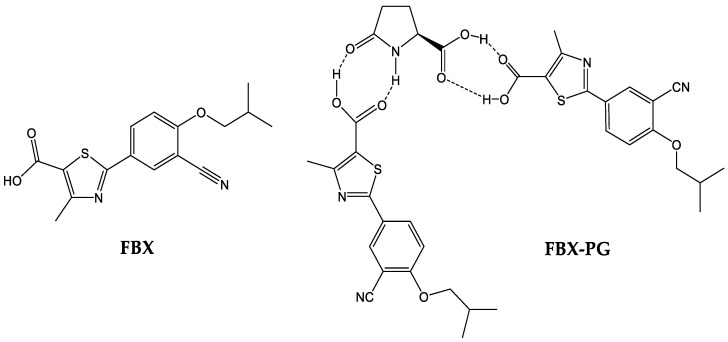
Structures of FBX and FBX-PG.

**Figure 2 pharmaceutics-15-02167-f002:**
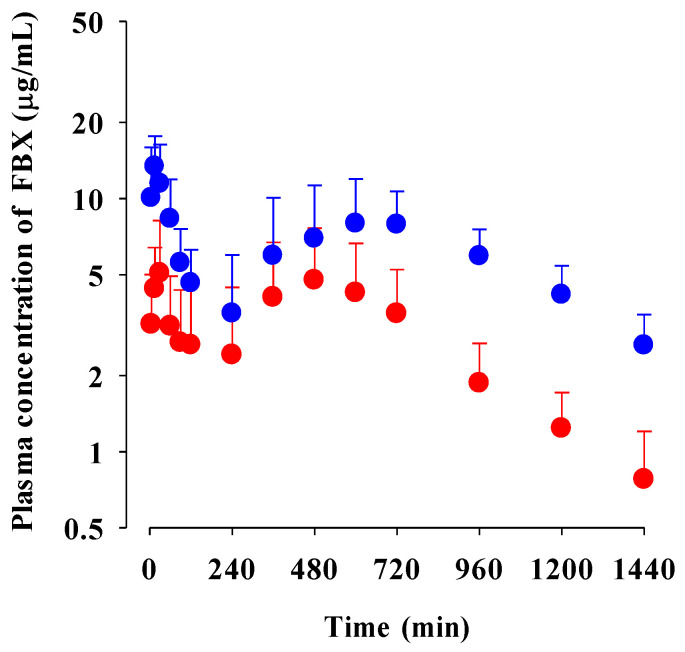
Mean plasma concentrations of FBX after the oral administration of FBX (●) and FBX-PG (●) at a dose of 50 mg/kg as FBX to rats. Error bars represent standard deviations. The *n* values of FBX and FBX-PG were 12 and 13, respectively.

**Figure 3 pharmaceutics-15-02167-f003:**
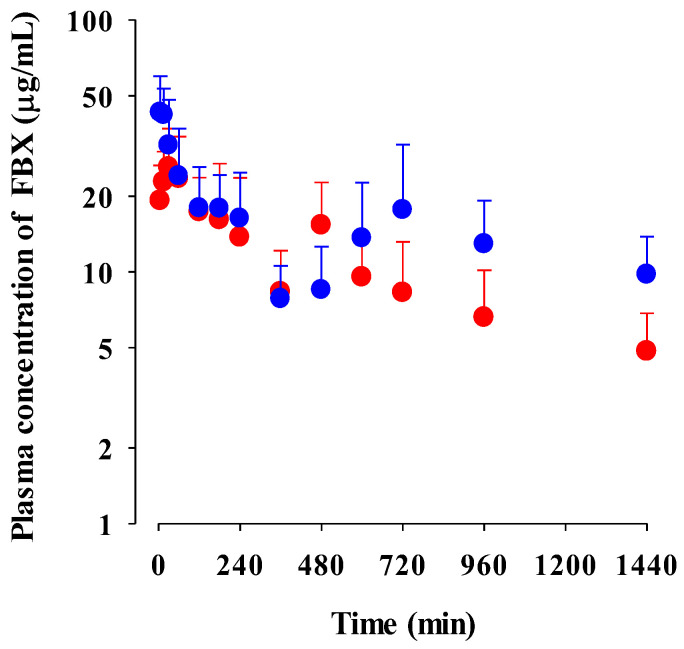
Mean plasma concentrations of FBX after the oral administration of FBX (●) and FBX-PG (●) at a dose of 50 mg/kg as FBX to mice. Error bars represent standard deviations. The *n* values of 22 and 15 represent the number of mice used in the FBX and FBX-PG-treated mice.

**Figure 4 pharmaceutics-15-02167-f004:**
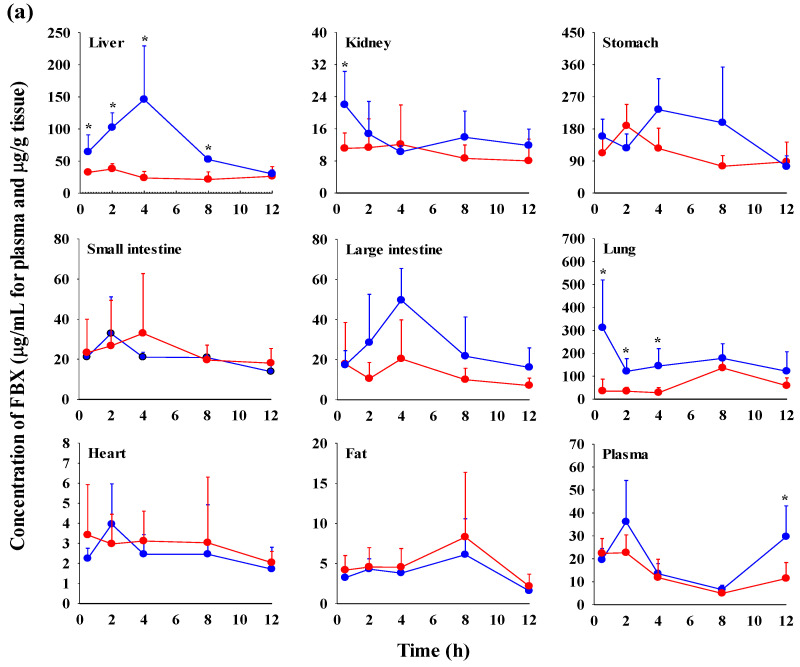
Mean concentrations (μg/mL for plasma and μg/g tissue) of FBX in plasma and tissue after the oral (50 mg/kg as FBX) administration of FBX (●) or FBX-PG (●) to mice (**a**). The T/P ratios of FBX after the oral (50 mg/kg as FBX) administration of FBX (■) or FBX-PG (■) to mice (**b**) are also shown. The dotted line in the graph shows that the T/P ratio is 1. Error bars represent standard deviations. The *n* values of 30 and 15 represent the number of mice used in the FBX and FBX-PG-treated mice * Statistically different (*p*-value < 0.05) from FBX-treated mice.

**Table 1 pharmaceutics-15-02167-t001:** Mean (±standard deviation) pharmacokinetic parameters of FBX after the oral administration of FBX or FBX-PG at a dose of 50 mg/kg as FBX to rats.

Parameters	FBX (*n* = 12)	FBX-PG (*n* = 13)
Body weight (g)	325 ± 37.4	330 ± 36.2
Terminal t_1/2_ (min)	434 ± 206	472 ± 207
*C*_max,1_ (μg/mL)	5.62 ± 2.86	14.6 ± 4.48 *
*T*_max,1_ (min) ^1^	30.0 (15–90)	15.0 (15–30) *
*C*_max,2_ (μg/mL)	5.63 ± 2.63	9.44 ± 3.27 *
*T*_max,2_ (min) ^1^	480 (120–960)	600 (480–960) *
AUC_0–1440min_ (μg min/mL)	4144 ± 1784	8251 ± 2566 *
AUC_0-inf_ (μg min/mL)	4824 ± 1960	10,149 ± 3164 *
V_z_/*F* (mL/kg)	7502 ± 4709	3508 ± 1567 *
CL/*F* (mL/min/kg)	12.5 ± 6.10	5.35 ± 1.58 *
*Ae*_0–24h_ (% of dose)	3.64 ± 1.25	3.69 ± 1.64
GI_24h_ (% of dose)	51.7 ± 15.7	40.8 ± 9.68 *
Relative *F*_0–1440min_ (%)		199
Relative *F* (%)		210

^1^ Data are median values (ranges). * Statistically different (*p*-value < 0.05) from FBX-treated rats.

**Table 2 pharmaceutics-15-02167-t002:** Mean pharmacokinetic parameters of FBX after the oral administration of FBX or FBX-PG at a dose of 50 mg/kg as FBX to mice.

Parameters	FBX (*n* = 22)	FBX-PG (*n* = 15)
Body weight (g)	35.3 ± 3.94	37.5 ± 3.62
Terminal t_1/2_ (min)	880	883
*C*_max,1_ (μg/mL)	26.0	42.9
*T*_max,1_ (min) ^1^	30.0	5.00
*C*_max,2_ (μg/mL)	15.3	17.6
*T*_max,2_ (min) ^1^	480	720
AUC_0–1440min_ (μg min/mL)	14,187	19,831
AUC_0-inf_ (μg min/mL)	20,324	32,253
V_z_/*F* (mL/kg)	3123	1974
CL/*F* (mL/min/kg)	2.46	1.55
Relative *F*_0–1440min_ (%)		140
Relative *F* (%)		159

Parameters are calculated using the average plasma concentrations from all mice in each group. The *n* values represent the number of mice used in the FBX and FBX-PG-treated mice. ^1^ Data are median values (ranges).

**Table 3 pharmaceutics-15-02167-t003:** Mean (±standard deviation) AUC_0–720min_ (μg min/g tissue) of FBX in tissues after oral administration of FBX or FBX-PG at a dose of 50 mg/kg as FBX to mice, respectively.

Tissue	FBX (*n* = 30)	FBX-PG (*n* = 15)
Liver	17,234 ± 5813	54,403 ± 10,551 *
Kidney	6961 ± 3954	9285 ± 619
Stomach	70,803 ± 19,964	114,976 ± 53,471
Small intestine	16,447 ± 8793	14,983 ± 2818
Large intestine	9434 ± 7230	18,978 ± 7719
Lung	48,314 ± 16,259	112,008 ± 19,080 *
Heart	1971 ± 841	1746 ± 738
Fat	3356 ± 1392	2834 ± 1302

* Statistically different (*p*-value < 0.05) from FBX-treated mice.

## Data Availability

Not applicable.

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
