# Peer review of "Evaluation of Pharmacokinetic Feasibility of Febuxostat/L-pyroglutamic Acid Cocrystals in Rats and Mice"

_pharmaceutics, 2023, doi:10.3390/pharmaceutics15082167_

Round 1
Reviewer 1 Report
The manuscript title "Evaluation of Pharmacokinetic feasibility of febuxostat/L-polyglutamic acid cocrystals in rats and mice" is an interesting research studies for the BCS class II drugs for showing the over-all improvement in the solubility of FBX and the improvement in oral bioavailability through pharmacokinetics studies and profile of FBX. The manuscript is well written which is interetsing and acceptable
1. There are room in the improvement of language and grammar throughout manuscript.
2. Why author shave selected both Rats and Mice animal species for bioavailability and pharmacokinetics profile of FBX and FBX PG formulation.
2.In abstract line 21-22 AUC of FBX is mentioned, please mentioned either 0-t or 0-∞ value.
3. The blood volume 150 µL (rats) and 120 µL (mice) were collected at 14 time point and 13 time points, see the guidelines (https://www.nc3rs.org.uk/3rs-resources/blood-sampling/blood-sampling-mouse#:~:text=How%20much%20blood%20does%20a,blood%20per%20kg%20of%20bodyweight.) for collection of the sample volume of blood as per the weight of the rats and mice. Check it
4. In Table 4. the concentration of FBX-PG mentioned in Liver tissue i.e. 54403± 10551, the SD values is to much high which is not acceptable.
5. Please add some more recent reference of the year 2022and 2023.
There are room in the improvement of language and grammar throughout manuscript.
Author Response
Response to Reviewer 1 Comments:
Point 1: There are room in the improvement of language and grammar throughout manuscript.
→ As you mentioned, the grammar and typos was corrected overall in the revised manuscript.
Point 2: Why author shave selected both Rats and Mice animal species for bioavailability and pharmacokinetics profile of FBX and FBX PG formulation.
→ Rodents, such as rats and mice, are the preferred species for drug development due to their anatomical, physiological, and genetic similarities to humans. In the field of pharmacokinetics, rats and mice have been widely employed as representative models due to their convenience (i.e., small body size and ease of maintenance), physiological resemblance to humans, and abundant genetic resources [Bryda et al (2013) in Mo Med, 110:207–211]. Rats have traditionally been used due to the advantage of allowing multiple blood sampling, which facilitates the calculation of pharmacokinetic parameters [Sivakrishman et al (2021) in Pharmacophore, 11:1–7]. Mice have gained attention as appealing alternatives for the following reasons: 1) smaller amounts of compounds are sufficient for pharmacokinetic studies, 2) various disease models have been developed using mice, and/or 3) mice are invaluable for animal-scale up studies [An et al (2012) in AAPS J, 14:352–364; Bryda et al (2013) in Mo Med, 110:207–211; Choi et al (2014) in Food Chem Toxicol, 66:140–146]. Due to the limited quantity of FBX-PG available for tissue distribution studies, mice were included in the study. Consequently, the enhanced bioavailability of FBX-PG was confirmed in both rat and mouse models, affiming the improved pharmaceutical performance of FBX-PG compared to FBX. Considering your comment, the reasons to employ rats and mice in this study were added in the discussion section (page 10, lines 326−338).
Point 3: In abstract line 21-22 AUC of FBX is mentioned, please mentioned either 0-t or 0-∞ value.
→ In abstract, the total area under the plasma concentration-time curve from time zero to the infinity was changed from AUC to AUC0-inf as you commented. Additionally, ‘AUC’ was changed to ‘AUC0-inf’ in the whole revised manuscript.
Point 4: The blood volume 150 µL (rats) and 120 µL (mice) were collected at 14 time point and 13 time points, see the guidelines (https://www.nc3rs.org.uk/3rs-resources/blood-sampling/blood-sampling-mouse#:~:text=How%20much%20blood%20does%20a,blood%20per%20kg%20of%20bodyweight.) for collection of the sample volume of blood as per the weight of the rats and mice. Check it.
→ The above references mentioned by the reviewer 1 [blood sampling: mouse in NC3Rs guidelines; Parasuraman et al (2010) in J Pharmacol Pharmacother, 1:87-93; Wolforth et al (2000) in Lab Anim (NY), 29:47-53] pertain to single point blood collection in the terminal stage, whereas multiple blood samplings in this study were also appropriate technique according to the guideline and reports [Golde et al (2005) in Lab Anim (NY), 34:39-43; Lee et al (2023) in Front Pharmacol, 14:1148155]. Therefore, the following responses address whether the blood collection technique is appropriate when multiple blood samplings are conducted in this study.
In blood collection technique for rats, the blood was collected through catheterization of the carotid artery. A single blood sampling of 0.15 mL and a total blood volume of 3 mL, which closely approximates 10% of the total blood volume per rat, are recommended in the guideline and report [blood sampling: rat in NC3Rs guidelines; Feng et al (2015) in J Vis Exp, 95:51881]. This study employed a single blood sampling of 0.15 mL at 14 blood sampling time points and a total blood sampling volume of 2.25 mL, adhering to the reported criteria [blood sampling: rat in NC3Rs guidelines; Feng et al (2015) in J Vis Exp, 95:51881]. In other words, the total blood sampling volume, 2.25 mL in this study, does not exceed the suggested value, 3.0 mL [blood sampling: rat in NC3Rs guidelines; Feng et al (2015) in J Vis Exp, 95:51881]. Furthremore, to prevent excessive blood loss, blood transfusion was administered between 12-h and 24-h blood sampling time points. This method is appropriate for multiple blood sampling in rats.
The blood collection technique in mice commonly involves heart puncture (also known as cardiac puncture) using a syringe equipped with a 19–27G needle [Hoggatt et al (2016) in Exp Hematol, 44:132-137; Parasuraman et al (2010) in J Pharmacol Pharmacother, 1:87-93; Sasidharan and Arunachalam (2021) 1st ed, Springer-Verlag, New York]. Guidelines recommend that a single blood sampling volume of 0.15 mL, a total blood sampling volume of 1.8 mL, and approximately three times of heart puncture in mouse with 20−25 g body weight [Diehl et al (2001) in J Appl Toxicol, 21:15-23; Golde et al (2005) in Lab Anim (NY), 34:39-43; Guidelines for rodent survival blood collection by FAU IACUC]. In this study, a 0.12 mL of blood was collected at each sampling time for three or four times, resulting in 0.36−0.48 mL of total blood was collected in each mouse with 25−30 g body weight. As each plasma concentration−time profile was constructed using data from three or four mice, blood was collected at three or four time points from each individual mouse, not at 13 time points. In other words, the total blood sampling volume, 0.36−0.48 mL in this study, does not exceed the suggested value, 1.8 mL [Parasuraman et al (2010) in J Pharmacol Pharmacother, 1:87-93; Wolforth et al (2000) in Lab Anim (NY), 29:47-53]. Furthermore, a 31G needle was employed to minimize damage to cardiac and pericardial tissues along the needle track, ensuring the mice’s survival for multiple blood collections. Thus, blood collection procedure adhered to the appropriate criteria for mice. The descriptions for appropriate blood sampling techniques in rats and mice were added in the discussion section of the revised manuscript (page 10, lines 348−369).
Point 5: In Table 4. the concentration of FBX-PG mentioned in Liver tissue i.e. 54403± 10551, the SD values is to much high which is not acceptable.
→ Table 4 was moved to Table 3 in the revised manuscript. The value you commented is the AUC720min of FBX, not its concentration, in FBX-PG group. The AUC720min of FBX indicates the area under the curve of FBX concentration versus time in the liver. Each individual data point in the following graph represents the FBX concentration at each sampling time in the the livers of each individual mouse. For the calculation of AUC720min, five data points were randomly selected from each sampling time and grouped together. In the statistical analysis using student t-test, the P value for AUC720min values between FBX and FBX-PG groups is 0.000206, despite a certain degree of variability being present. When calculating the AUC720min values of FBX using the mean concentration of FBX at each sampling time in both groups, the AUC720min values of FBX were 18,369 and 54,953 μg min/mL in FBX and FBX-PG groups, respectively. These values imply a significant increase in the AUC720min of FBX in FBX-PG group compared to that in FBX group.
Point 6: Please add some more recent reference of the year 2022 and 2023.
→ As you mentioned, recent references of the year 2022 and 2023 were reflected in the revised manuscript, which was added to reference lists.

Reviewer 2 Report
The study evaluation of pharmacokinetic feasibility of febuxostat/L-pyroglutamic acid cocrystals in rats and mice. The results suggest the FBX-PG has a suitable pharmacokinetic profile. The results of the paper have reference value for improving the biological activity of Febuxostat.
Some points:
The introduction section could add some recent progress on improving the solubility of this drug.
Line 225, Figure 2. “was” to “were”
Line 264. in the lung
Line 314. In a pharmacokinetic study
Line 320. In the evaluation of
Line 347. the second peak
Author Response
Response to Reviewer 2 Comments:
Point 1: The introduction section could add some recent progress on improving the solubility of this drug.
→ Various formulations of FBX aimed to improve its solubility have been introduced including salt formations [Han et al (2016) in Asian J Pharm Sci, 11:715–721], solid dispersion formations [Sohn and Choi (2023) in Saudi Pharm J, 31:101724] and co-crystal formations [Gao and Zhang (2021) in J Chem, 2021:1–8; Kang et al (2016) in Molecular structure, 15:480–486]. In the circumstances that co-crystal formation has emerged as one of notable crystal engineering techniques to change the pharmaceutical performances, such as solublility, dissolution rate, bioavailability, compacitibility, and physical stability, of the API by controlling the solid-state properties of the API [Good and Rodriguez-Hornerdo (2008) in Cryst Growth Des, 9:2252–2264; Remenar et al (2003) in J Am Chem Soc, 125: 8456-8457; Jung et al (2010) in J Pharm Pharmacol, 62: 1560-1568; Rahman (2011) AAPS PharmSciTech, 12:693-704; Karki et al (2009) in Advanced materials, 21:3905-3909; Trask et al (2006) in Int J Pharm, 320:114-123], various co-crystals of FBX have been developed such as co-crystals of with isonicotinamide, arginine, or pyridine [Gao and Zhang (2021) in J Chem, 2021:1–8; Kang et al (2016) in Molecular structure, 15:480–486]. Also six distinct coformers, such as 1-hydroxy 2-naphthoic acid, 4-hydroxy benzoic acid, salicylic acid, 5-nitro isophthalic acid, isonicotinamide, and picolinamide, all of which were shown to enhance the solublity of FBX, but these co-crystals of FBX exhibited instability in the buffered solutions of pH 4–7 [Jagia et al (2021) in AAPS PharmSciTech, 23:43]. In contrast, it was reported the improved solubility of FBX-PG in both deionized water and buffers reflecting the pH conditions of the gastrointestial tract different from other co-crystals of FBX [An et al (2017) in Crystals, 7:365; Jagia et al (2021) in AAPS PharmSciTech, 23:43]. Considering that the solubility and stability of an orally administered drug in the different pH conditions of gastrointestinal tract influence importantly the absorption and bioavailability [Abuhelwa et al (2016) in Eur J Pharm Biopharm, 112:234–248], FBX-PG has an advantage for improving F.
Point 2: Line 225, Figure 2. “was” to “were”
→ Page 6, line 255, Figure 2: As you mentioned, “was” was modificated to “were” in the legend of Figure 2.
Point 3: Line 264. in the lung
→ Page 8, line 293: As you mentioned, “in lung” was changed to “in the lung” in the revised manuscript. In addition, other sentences were also corrected.
Point 4: Line 314. In a pharmacokinetic study
→ Page 10, line 320: As you mentioned, “In pharmacokinetic study” was changed to “In a pharmacokinetic study” in the revised manuscript.
Point 5: Line 320. In the evaluation of
→ Page 11, line 373: As you mentioned, “In evaluation of” was changed to “In the evaluation of” in the revised manuscript.
Point 6: Line 347. the second peak
→ Page 6, line 239; page 7, line 269; page 11, lines 413-415; page 12, lines 422-425: As you mentioned, “the second peaks” was modificated to "the second peak" in the revised manuscript. In addition, other sentences were also corrected.

Reviewer 3 Report
The manuscript presents scientifically significant in vivo study on the pharmacokinetics profile and feasibility of applying febuxostat/L- 2-pyroglutamic acid cocrystals in rats and mice and the advantages of the cocrystals over febuxostat alone. However, the following inaccuracies and weak issues were detected:
1. All abbreviations used have to be given in full at first mention – e.g. AUC in the Abstract.
2. The introduction section has to be supplemented with novel studies on the applicability and advantages of other formulations offering better bioavailability of the tested physiologically active compound, as well as to emphasize on the pros of the use of L-pyroglutamic acid over other co-crystal former molecules. (e.g. Jagia M, Kale DP, Bansal AK, Patel S. Novel Co-crystals and Eutectics of Febuxostat: Characterization, Mechanism of Formation, and Improved Dissolution. AAPS PharmSciTech. 2021 Dec 29;23(1):43. doi: 10.1208/s12249-021-02182-9. )
3. The method for preparation of febuxostat/L-2-pyroglutamic acid cocrystals has to be described in brief.
4. The software - Swiss ADME program, applied for prediction of the physicochemical properties of FBX and FBX-PG has to be mentioned either in a separate section or in subsection Statistical analyses in Material and Methods.
5. The breed of the mice applied has to be specified.
6. As the data in Table 1 are not obtained in the present study, it is not correct to rewrite them but just to cite the reference. Similar study was conducted in 2022 and the experimental data have to b e compared and cited: Haq N, Alghaith AF, Alshehri S, Shakeel F. Solubility and Thermodynamic Data of Febuxostat in Various Mono Solvents at Different Temperatures. Molecules. 2022 Jun 23;27(13):4043. doi: 10.3390/molecules27134043.
7. Subsection 2.5. – Were the mice under anaesthesia when blood was collected via heart puncture?
8. In the Materials and Methods section, the experimental groups of rats and mice have to be clearly described – number of animals, purpose of the group (tissue distribution analyses, blood analyses, etc) as when reading the experimental design, one gets confused.
9. Subsection 2.6. – How were the animals euthanized?
10. Although identical doses of the two formulations were administered to the experimental animals, probably the real dose of the active substance in the FBX-PG co-crystals was lower due to the presence of an additional substance and the higher molar mass. This fact has to be taken into account when presenting the experimental results and their discussion.
11. The manuscript would be more attractive for the reader if colour figures are applied.
12. A Conclusions section is missing and has to be added.
Author Response
Response to Reviewer 3 Comments:
Point 1: All abbreviations used have to be given in full at first mention – e.g. AUC in the Abstract.
→ As you mentioned, all abbreviations in full at first mention were clearly added in the revised manuscript.
Point 2: The introduction section has to be supplemented with novel studies on the applicability and advantages of other formulations offering better bioavailability of the tested physiologically active compound, as well as to emphasize on the pros of the use of L-pyroglutamic acid over other co-crystal former molecules. (e.g. Jagia M, Kale DP, Bansal AK, Patel S. Novel Co-crystals and Eutectics of Febuxostat: Characterization, Mechanism of Formation, and Improved Dissolution. AAPS PharmSciTech. 2021 Dec 29;23(1):43. doi: 10.1208/s12249-021-02182-9.)
→ As the bioavailiablity (F) of a drug is one of determinants for the therapeutic effect [Rekdal et al (2018) in Syst Rev Pharm, 9:55–57], the formulations offering better F and better physiochemical properties related to improving F have the advantages [Ameta et al (2023) in Colloids Interfaces, 7:16; Gupta et al (2013) in ISRN Pharm, 2013:848043; Hichey et al (2007) in Eur J Pharm Biopharm, 67:112–119]. Co-crystal formation has emerged as one of notable crystal engineering techniques to change the pharmaceutical performances, such as solublility, dissolution rate, bioavailability, compacitibility, and physical stability, of the API by controlling the solid-state properties of the API [Good and Rodriguez-Hornerdo (2008) in Cryst Growth Des, 9:2252–2264; Jung et al (2010) in J Pharm Pharmacol, 62: 1560-1568; Karki et al (2009) in Advanced materials, 21:3905-3909; Rahman (2011) in AAPS PharmSciTech, 12:693-704; Remenar et al (2003) in J Am Chem Soc, 125:8456-8457; Trask et al (2006) in Int J Pharm, 320:114-123]. Although FBX form-A is still used in drug formulation, the co-crystallization of FBX has been diversely tried due to the low aqueous solubility and slow absorption rate of FBX [Amin et al (2019) in J Pharm Investig, 50:399–411; An et al (2017) in Crystals, 7:365; Banerjee et al (2005) in Cryst Growth Des 5:2299−2309; Basavoju et al (2006) in Cryst Growth Des 6:2699−2708; Byrn et al (1999) 2nd ed. West lafayette, Indiana; Good and Rodríguez-Hornedo (2009) in Cryst Growth Des, 9:2252−2264; Maddileti et al (2013) in Cryst Growth Des, 13:3188–3196; McNamara et al (2006) in Pharm Res, 23:1888−1897; Remenar et al (2003) in J Am Chem Soc, 125:8456−8457; Sanphui (2011a) in Cryst Growth Des, 11:4135−4145; Sanphui et al (2011b) in Chem Commun (Camb), 47:5013−5015]. Until now, several studies reported that the low aqueous solubility of FBX was improved through forming the co-crystals [Gao and Zhang (2021) in J Chem, 2021:1–8; Jagia et al (2021) in AAPS PharmSciTech, 23:43; Kang et al (2016) in Molecular structure, 15:480–486], but the solubility and stability of FBX cocrystals in various pH environments have been still issued [Jagia et al (2019) in Mol Pharm, 4:4610–4620]. Since the solubility and stability of the drug in the pH conditions of gastrointestinal tract is an important factor to affect absorption and bioavailability [Abuhelwa et al (2016) in Eur J Pharm Biopharm, 112:234–248; Jagia et al (2021) in AAPS PharmSciTech, 23:43], the improved solubiltiy and stability of FBX-PG in simulated gastrointestinal pH conditions proposed that FBX-PG has an advantage for the potential of improving F compared to FBX itself [An et al (2017) in Crystals, 7:365] and other co-crystals of FBX [Jagia et al (2021) in AAPS PharmSciTech, 23:43]. These sentences were also relfected in the revised manuscript (pages 1-2, lines 41-66).
Point 3: The method for preparation of febuxostat/L-2-pyroglutamic acid cocrystals has to be described in brief.
→ The preparation method of febuxostat/L-2-pyroglutamic acid cocrystal (FBX-PG) was published by An et al. (2017). As the manufactured FBX-PG was provided and used in this study, a brief summary for the preparation method of FBX-PG [An et al (2017) in Crystals, 7:365; Ryu (2015) K.R. Patent 10-1501253] was added in the subsection of “2.1 Materials” considering your advice as follows (page 2, lines 73-77).
Point 4: The software - Swiss ADME program, applied for prediction of the physicochemical properties of FBX and FBX-PG has to be mentioned either in a separate section or in subsection Statistical analyses in Material and Methods.
→ I agree to your comment, and Table 1 was deleted. The physicochemical properties of FBX and FBX-PG were discussed in the discussion section of the revised manuscript.
Point 5: The breed of the mice applied has to be specified.
→ The breed of rats and mice used in this study were Institute of Cancer Research (ICR) mice and Sprague-Dawley (SD) rats, respectively. This information has been already described in the “2.2. Animals” section of the original manscript (page 2, lines 84-86).
Point 6: As the data in Table 1 are not obtained in the present study, it is not correct to rewrite them but just to cite the reference. Similar study was conducted in 2022 and the experimental data have to be compared and cited: Haq N, Alghaith AF, Alshehri S, Shakeel F. Solubility and Thermodynamic Data of Febuxostat in Various Mono Solvents at Different Temperatures. Molecules. 2022 Jun 23;27(13):4043. doi: 10.3390/molecules27134043.
→ I agree to your comment, and Table 1 was deleted. The physicochemical properties of FBX and FBX-PG were discussed in the discussion section of the revised manuscript. Also Haq et al. (2022) metioned in your comment reported the water solubility of FBX, not FBX-PG, which was refered in the revised manuscirpt (page 2, line 48).
Point 7: Subsection 2.5. – Were the mice under anesthesia when blood was collected via heart puncture?
→ Anesthesia is recommended for blood sampling using cardiac puncture [Parasuraman et al (2010) in J Pharmacol Pharmacother, 1:87-93]. Thus, the blood samplings via heart puncture in mice were performed under anestheria according to the protocol approved (IACUC-2022-12) and previous reported methods [Diehl et al (2001) in J Appl Toxicol, 21:15−23; Golde et al (2005) in Lab Anim (NY), 34:39−43; Lee et al (2023) in Front Pharmacol, 14:1148155]. Anesthesia was conducted by i.p. injection of 0.05 mL per kg composing 3:1 mixture of zoletil (i.e., tiletamine 125 mg + zolazepam 125 mg/5 mL) and rompun (xylazine HCl 23.3 mg/5 mL) before heart puncture. The blood samplings via heart puncture in mice were performed as described in the subsections of “2.5. Pharmacokinetics of FBX after oral administration of FBX or FBX-PG in mice” and “2.6. Tissue distribution of FBX after oral administration of FBX or FBX-PG to mice” in the revised manuscript. As you mentioned, the descrrption about anesthesia was added as following (page 4, lines 158-161; page 4, lines 176-179).
Point 8: In the Materials and Methods section, the experimental groups of rats and mice have to be clearly described – number of animals, purpose of the group (tissue distribution analyses, blood analyses, etc) as when reading the experimental design, one gets confused.
→ As you mentioned, the experimenatl groups of rats and mice were more clearly described in several sections:
Page 4, lines 131-132 in subsection of “2.4. Pharmacokinetics of FBX after oral administration of FBX or FBX-PG to rats”: On the day of the experiment, the rats were randomly divided into two groups, such as FBX (n = 12) and FBX-PG (n=13) groups, respectively.
Page 4, lines 157-158 in subsection of “2.5. Pharmacokinetics of FBX after oral administration of FBX or FBX-PG in mice”: On the day of the experiment, mice were randomly divided into two groups, such as FBX (n = 22) and FBX-PG (n = 15) groups, respectively.
Page 4, lines 175-176 in subsection of “2.6. Tissue distribution of FBX after oral administration of FBX or FBX-PG to mice”: On the experiment day, mice were randomly divided into two groups of mice, such as FBX (n = 30) and FBX-PG (n = 15) groups, respectively.
Point 9: Subsection 2.6. – How were the animals euthanized?
→ In subsection 2.6., each mouse was euthanized at each individual sampling time point. A feasible volume of blood was collected via heart puncture and the portal vein perfusion with 0.9% NaCl solution was performed to remove all blood from the body. This description has been already written in the original manuscript (as described in subsection 2.6; page 5, lines 187-189).
Point 10: Although identical doses of the two formulations were administered to the experimental animals, probably the real dose of the active substance in the FBX-PG co-crystals was lower due to the presence of an additional substance and the higher molar mass. This fact has to be taken into account when presenting the experimental results and their discussion.
→ The statement “..at a dose of 50 mg/kg as FBX..” indicates that the dose was standarized based on FBX. FBX-PG (761.85 g/mol) consists of a 1:2 ratio of FBX (316.37 g/mol) and L-Pyroglutamic acid (129.04 g/mol). Thus, the dose of FBX-PG was adjusted by considering the molar concentration of FBX (316.37 g/mol) and L-Pyroglutamic acid (129.04 g/mol) within FBX-PG, resulting in an equivalent dose of FBX within FBX-PG. These statements were added the revised manuscript (page 4, lines 138-141; page 4, lines 163-165; page 5, lines 181-184).
Point 11: The manuscript would be more attractive for the reader if color figures are applied.
→ As you recommended, figure symbols in Figures 2, 3, and 4 were changed the color.
Point 12: A Conclusions section is missing and has to be added.
→ As you mentioned, a conclusion section was added in the revised manuscript.

Reviewer 4 Report
In this study, the pharmacokinetic feasibility of Febuxostat/L-pyroglutamic acid cocrystal (FBX-PG), a novel cocrystals of FBX, was evaluated by comparing FBX-PG with FBX in rats and mice. Although there are not obvious innovations on the methodology, this study has obtained some interesting results. This manuscript may be accepted for publication after revisions. Followings are some suggestions for revisions.
1, The introduction section can be improved. Some exact examples of cocrystals and recent studies on the in vivo pharmacokinetic property of cocrystals can be introduced. The key innovations of this study can be emphasized in the last part of introduction section.
2, Why both the rats and mice were chosen for the Pharmacokinetics study?
3, Why only the tissue distribution in mice was investigated? As both the rats and mice were chosen for the Pharmacokinetics study.
4, Will the FBX-PG affect the curative effect of FBX? Are there any related studies?
5, The discussion section can be improved. Some paragraphs can be combined. The text can be written in a more logic way.
6, It is suggested to add a conclusion section.
7, Please check the units. For example, the “12,000 rpm” and “9,000 rpm” can be converted to centrifugal force “g”.
Author Response
Response to Reviewer 4 Comments:
Point 1: The introduction section can be improved. Some exact examples of cocrystals and recent studies on the in vivo pharmacokinetic property of cocrystals can be introduced. The key innovations of this study can be emphasized in the last part of introduction section.
→ As co-crystal formation has emerged as one of notable crystal engineering techniques to change the pharmaceutical performances, such as solublility, dissolution rate, bioavailability, compacitibility and physical stability, of the API by controlling the solid-state properties of the API [Good and Rodriguez-Hornerdo (2008) in Cryst Growth Des, 9:2252–2264; Remenar et al (2003) in J Am Chem Soc, 125: 8456-8457; Jung et al (2010) in J Pharm Pharmacol, 62: 1560-1568; Rahman (2011) AAPS PharmSciTech, 12:693-704; Karki et al (2009) in Advanced materials, 21:3905-3909; Trask et al (2006) in Int J Pharm, 320:114-123], co-crystals are known to be superior to conventional formulations especially in in vivo pharmacokinetic properties such as absorption and F. For example, the AUC values of the APIs were increased by 2.7, 1.95, and 24 times in co-crystals of apixaban with oxalic acid, indomethacin with saccharin, and apixaban with quercetin, respectively, compared to the corresponding value of the API itself [Chen et al (2016) in Cryst Growth Des, 16:2923-2930; Jung et al (2010) in J Pharm Pharmacol, 62:1560-1568; Zhang et al (2021) in Molecules, 26:2677]. In case of FBX cocrystals, the solubility and thermodynamical stability of FBX-PG in solid states were improved through the regulation of PG among pharmaceutically usable salts. Moreover, the solubility was improved in both deionized water and buffers reflecting the pH conditions of the gastrointestial tract [An et al (2017) in Crystals, 7:365] in contrast to other co-crystals of FBX showing the enhanced solubility being unstable in the buffer with pH 4-7 [Jagia et al (2021) in AAPS PharmSciTech, 23:43]. Considering that the solubility and stability of an orally administered drug in the different pH conditions of gastrointestinal tract importantly influence the absorption and bioavailability, the better in vivo pharmacokinetic feasibility of FBX-PG is expected [An et al (2017) in Crystals, 7:365]. Nevertheless, in vivo pharmacokinetic property of FBX after administration of FBX-PG has not been evaluated yet. Threfore, in vivo pharmacokinetic feasibility of FBX-PG was compared to FBX in rats and mice. As you commented, the sentence was added in the introduction section of the revised manuscript (pages 1-2, lines 41-66).
Point 2: Why both the rats and mice were chosen for the Pharmacokinetics study?
→ Rodents, such as rats and mice, are the preferred species for drug development due to their anatomical, physiological, and genetic similarities to humans. In the field of pharmacokinetics, rats and mice have been widely employed as representative models due to their convenience (i.e., small body size and ease of maintenance), physiological resemblance to humans, and abundant genetic resources [Bryda et al (2013) in Mo Med, 110:207–211]. Rat models have traditionally been used because rats allow for multiple blood sampling, which facilitates the calculation of pharmacokinetic parameters [Sivakrishman et al (2021) in Pharmacophore, 11:1–7]. Mouse models have gained attention as appealing alternatives for the following reasons: 1) smaller amounts of compounds are sufficient for pharmacokinetic studies, 2) various disease models have been developed using mice, and/or 3) mice are invaluable for animal-scale up studies [An et al (2012) in AAPS J, 14:352–364; Bryda et al (2013) in Mo Med, 110:207–211; Choi et al (2014) in Food Chem Toxicol, 66:140–146]. Consequently, the enhanced bioavailability of FBX-PG was confirmed in both rat and mouse models, affiming the improved pharmaceutical performance of FBX-PG compared to FBX. This description was added in the discussion section of the revised manuscript (page 10, lines 326−338).
Point 3: Why only the tissue distribution in mice was investigated? As both the rats and mice were chosen for the Pharmacokinetics study.
→ Due to the limited quantity of FBX-PG available for tissue distribution studies, mice were only employed. Mouse models have gained attention as appealing alternatives for the following reasons: 1) smaller amounts of compounds are sufficient for pharmacokinetic studies, 2) various disease models have been developed using mice, and/or 3) mice are invaluable for animal-scale up studies, not unlike rats [An et al (2012) in AAPS J, 14:352–364; Bryda et al (2013) in Mo Med, 110:207–211; Choi et al (2014) in Food Chem Toxicol, 66:140–146].
Although tissue distribution study was conducted using only mice, the increased bioavailability of FBX-PG was confirmed in both rat and mouse models. Rats and mice are rodents, and both of rats and mice have been generally used as representative models in pharmacokinetics research due to convenience (i.e., small body size and ease of maintenance), similarity to human physiology, and abundant genetic resources [Bryda et al (2013) in Mo Med, 110:207–211]. The homology between rats and mice is quite high genetically, with over 90% similarity in protein sequences [Lund and Holmverg (2006) in Cell Tissue Res, 324:35–40]. As a result, the increased bioavailability of FBX-PG was confirmed in both rat and mouse models, affiming the improved pharmaceutical performance of FBX-PG compared to FBX. This description was added in the discussion section of the revised manuscript (page 10, lines 326−338).
Point 4: Will the FBX-PG affect the curative effect of FBX? Are there any related studies?
→ As described in the first paragraph in the discussion section (page 10, lines 320-338), the drug exposure at appropriate concentrations during a sufficient duration is a critical factor in achieving the expected efficacy. In particular, drug exposure in pharmacokinetic characteristics of a drug play an important role in determining the expected drug response over time (i.e., efficacy and/or toxicity) [Cook et al (2014) in Nat Rev Drug Discov, 13:419–431; Lin (2006) in Curr Drug Metab, 7:39–65; Rekdal et al (2018) in Syst Rev Pharm, 9:55–57]. Owing to the improved solubility and stability of FBX-PG, the AUC representing systemic exposure was increased and consequently the F was enhanced following FBX-PG adminstration to rats and mice compared to FBX administration. Similar to the previously reported cases [Rekdal et al (2018) in Syst Rev Pharm, 9:55–57], the increased systemic exposure and F values of FBX in FBX-PG group may affect the curative effect of FBX. Although the quantitiative change of curative effect of FBX-PG has not been reported yet, the physiochemical properties, such as solubility and stability, of FBX resulted in improving F, which is belived to affect the curative effect of FBX following FBX-PG administration.
Point 5: The discussion section can be improved. Some paragraphs can be combined. The text can be written in a more logic way.
→ As you mentioned, the discussion section was revised.
Point 6: It is suggested to add a conclusion section.
→ As you mentioned, conclusions section was added in the revised manuscript.
Point 7: Please check the units. For example, the “12,000 rpm” and “9,000 rpm” can be converted to centrifugal force “g”.
→ As you commented, “12,000 rpm” and “9,000 rpm” were converted to “13,523 x g” and “7,607 x g” in the revised manuscript (page 3, line 122; page 5, lines 187–190).

Round 2
Reviewer 3 Report
The manuscript is acceptable for publication in its present form.
Author Response
We appreciate your comments for our manuscript.
Reviewer 4 Report
The manuscript has been carefully revised according to the suggestions. This manuscript may be accepted for publication after minor revisions.
1, The quality (resolution) of figure can be further improved.
2, The limitations of this study can be emphasized in the conclusion section.
Author Response
Response to Reviewer 4 Comments:
Point 1: The quality (resolution) of figure can be further improved.
→ As you mentioned, the resolution of figure (especially, figure 1) was improved in the revised manuscript. All figures in the revised manuscript were revised to increase the resolution as following: Figure 1 (3239 x 1525 pixels, 330 dpi), Figure 2 and 3 (1613 x 1690 pixels, 330 dpi), Figure 4 (2329 x 3778 pixels, 330 dpi). All figures were followed by the author instruction (the figure must be adjusted to at least 1000 pixels (width/height) or at least 300 dpi resolution).
Point 2: The limitations of this study can be emphasized in the conclusion section.
→ As you mentioned, the limitaions of this study were added in the conclusion section as follows:
5. Conclusion
It can be concluded that the FBX-PG administration significantly increased the AUC0-inf and F values of FBX in rats and mice. Despite the high plasma protein binding value of FBX, the tissue distribution of FBX to livers and lungs exhibited the significant increases after FBX-PG administration compared to FBX. These findings suggest that the potential of FBX-PG as a promising formulation to enhance FBX exposure in both plasma and tissues, thereby possibly improving therapeutic efficacy. However, further studies are required to elucidate the changes between exposure and therapeutic efficacy of FBX after FBX-PG administration. Additionally, exploring underlying mechanisms such as entero-hepatic circulation provide valuable insights to explain the systemic exposure of FBX beyond the increased absorption observed with oral administration of FBX-PG.